# Revolutionizing Chinese sentiment analysis: A knowledge-driven approach with multi-granularity semantic features

Ping He ⓘ *

Changsha Institute of Technology, Changsha, China

* 240670821@qq.com

## Abstract

In recent years, there has been significant progress in Chinese text sentiment analysis research. However, few studies have investigated the differences between languages, the effectiveness of domain knowledge, and the requirements of downstream tasks. Considering the uniqueness of Chinese text and the practical needs of sentiment analysis, this study aims to address these gaps. To achieve this, we propose a method that deeply integrates the knowledge vector obtained from the emotional knowledge triplets using the TransE model with feature vectors from models like BiGRU and attention mechanisms. We also introduce radical features and emotional part of speech features based on the characteristics of characters and words. In addition, we propose a collaborative approach that integrates characters, words, radicals, and multi-granularity semantic features such as part of speech. Our approach, as evidenced by the Douban Film Review dataset and the NLPECC dataset, proficiently leverages emotional insights alongside nuanced linguistic elements, significantly bolstering the accuracy of sentiment detection in Chinese texts. The method achieved F1-score of 89.23% and 84.84%, respectively, underscoring its efficacy in the realm of Chinese sentiment analysis.

## Introduction

The domain of sentiment analysis within the Chinese language has attracted considerable research interest and has made significant strides. Despite these advancements, the majority of studies have concentrated on the "data-driven approach", employing deep learning algorithms to decode the profound semantic content embedded in Chinese textual data. Yet, this methodology encounters difficulties in addressing the phenomenon of word polysemy, leading to challenges such as unclear semantic interpretations and a lack of emotional expressiveness. Furthermore, the intrinsic linguistic differences between Chinese and English pose a barrier to the straightforward application of sentiment analysis methodologies developed for English to Chinese texts, complicating the task of sentiment analysis in Chinese.

**Citation:** He P (2025) Revolutionizing Chinese sentiment analysis: A knowledge-driven approach with multi-granularity semantic features. PLoS One 20(7): e0325428. https://doi.org/10.1371/journal.pone.0325428

**Data availability statement:** All relevant data are within the manuscript and its Supporting Information files.

**Funding:** This work was supported by Joint Funding Project for Foreign Language Research of Hunan Provincial Social Science Foundation, grant number 24WLH30, awarded to PH, byt the Higher Education Scientific Research Project, grant number 23WYJ0307, awarded to PH, by the Teaching Reform Project of Hunan Provincial Department of Education, grant number HNJG20231251, awarded to JC. The Joint Funding Project for Foreign Language Research of Hunan Provincial Social Science Foundation played a role in study design, data collection and analysis and decision to publish. The Higher Education Scientific Research Project played a role in study design and preparation of the manuscript. The Teaching Reform Project of Hunan Provincial Department of Education played a role in study design and preparation of the manuscript.

**Competing interests:** For a paper with no competing interests: The authors have declared that no competing interests exist. For a paper with competing interests: I have read the journal's policy and the authors of this manuscript have the following competing interests: [Specifically describe the competing interests here, such as funding sources, financial interests in related companies, patents, etc.].

To surmount the aforementioned hurdles, our research embarks on a dual-strategy initiative, bridging the gaps between "knowledge" and "data" dimensions. Within the "knowledge" domain, we harness pre-existing insights encapsulated in an affective knowledge graph, adeptly navigating the complexities of word meanings across varied contexts. Concurrently, on the "data" front, we concentrate on the unique structural nuances of Chinese script and the tangible demands of sentiment analysis. We endeavor to enrich the emotional spectrum by tapping into the intrinsic qualities of characters and words, such as radicals and emotional part of speech markers. Our proposed methodology integrates these with the profound insights gleaned from advanced models, including the bidirectional gated recurrent unit (BiGRU) and sophisticated attention mechanisms, to forge a sentiment analysis framework that seamlessly amalgamates multi-level semantic elements—ranging from characters to words and beyond—in a knowledge- and data-driven synergy.

### Contributions of the study: Innovations in methodology and multidisciplinary approaches to chinese sentiment analysis

Our research presents three pivotal contributions to the field. Initially, we tackle the distinctive attributes of Chinese textual data and the specific demands of sentiment analysis through the strategic use of part-of-speech classifications for Chinese radicals and lexicon, thereby enhancing the semantic comprehension of the text. This targeted approach delves into the intricacies of sentiment analysis within the Chinese language, offering novel perspectives and advancements.

Furthermore, our work synthesizes state-of-the-art theoretical frameworks and analytical techniques from a spectrum of disciplines, encompassing computational intelligence, linguistic studies, information technology, and the realm of artificial intelligence. This comprehensive, cross-disciplinary strategy not only equips us with a robust arsenal of innovative methodologies but also facilitates the generation of solutions that are both creative and effective.

Lastly, we pioneer the adoption of the "knowledge+data" research paradigm in the arena of Chinese sentiment analysis. This novel approach amalgamates emotional knowledge vectors with the feature vectors derived from sophisticated deep learning models, culminating in a sentiment analysis methodology that harnesses a spectrum of semantic features ranging from characters to words, radicals, and parts of speech. Such an integrative strategy not only bolsters the analytical prowess of Chinese sentiment analysis but also contributes to the enrichment and evolution of its theoretical and methodological underpinnings.

In summary, our research introduces a novel suite of methodologies and a multidisciplinary lens to the domain of Chinese sentiment analysis. This innovative framework paves the way for fresh perspectives and deeper understanding, propelling the field forward with its unique contributions:

**Novel Feature Integration:** Systematic incorporation of radical features (e.g., 忄, 心) and emotional part-of-speech features (verbs/adjectives) into Chinese sentiment analysis, addressing linguistic uniqueness.

**Knowledge-Data Synergy:** A hybrid framework fusing emotional knowledge vectors (via TransE) with deep learning features (BiGRU + multi-head attention), enabling multi-granularity semantic fusion.

**Cross-Disciplinary Methodology:** Integration of computational linguistics, AI, and knowledge graphs, validated on large-scale datasets (F1: 89.23% on Douban, 84.84% on NLPECC).

## Related works

The landscape of Chinese text sentiment analysis is segmented into three principal methodologies: sentiment analysis propelled by knowledge constructs, data-centric approaches, and hybrid models that amalgamate knowledge with data.

Sentiment analysis research propelled by knowledge primarily concentrates on devising knowledge graphs, distilling salient features from these constructs, and formulating rules for sentiment detection. Yet, this avenue of research has its constraints. Xu Zhihong and colleagues introduced a semi-automated strategy for mining insights from film critiques, thereby fortifying the efficacy of film and TV show recommendations [1]. Their approach encompassed the creation of a knowledge graph that encapsulated various film review attributes. It entailed a series of steps including the preprocessing of review texts, syntactic analysis, compilation of sentiment lexicons, and data annotation, all converging to establish a framework for knowledge extraction. By integrating lexicons with quantitative clustering techniques, they derived a structured repository of film review insights. These insights were subsequently merged with film ontology, culminating in an enriched knowledge graph tailored for sentiment analysis. Despite its simplicity in concept, knowledge-driven research often overlooks the contextual intricacies of language, neglecting to fully harness the depth of semantic content and thus restricting its broad applicability.

Conversely, data-driven approaches to sentiment analysis hinge on sophisticated deep learning architectures and expansive datasets, enabling the extraction of salient features and the discernment of sentiment patterns. These methodologies harness the prowess of neural networks to delineate intricate associations and bolster the models' capacity for generalization. Nonetheless, they sometimes grapple with challenges pertaining to the multiple meanings of words and the nuanced conveyance of emotions.

Acknowledging the inherent constraints of both the knowledge-driven and data-driven paradigms, the hybrid approach in sentiment analysis endeavors to amalgamate their respective strengths. This strategy integrates the structured insights from knowledge graphs with the robust predictive capabilities of deep learning models, aspiring to refine sentiment analysis through an enriched tapestry of contextual insights and profound semantic comprehension.

To encapsulate, the realm of Chinese text sentiment analysis is traversed by three dominant methodologies: the knowledge-driven, data-driven, and their hybrid iteration. While each brings its unique set of merits and demerits to the table, there is an imperative for ongoing research to forge more efficacious and holistic approaches to sentiment analysis.

Within the purview of data-driven sentiment analysis, two principal methodologies are predominantly employed: the conventional approach and the avant-garde deep learning techniques. The conventional approach often relies on sentiment lexicons or machine learning algorithms to perform sentiment analysis [2,3]. Despite delivering adequate outcomes, these methods necessitate considerable investment in terms of labor, resources, and the meticulous crafting of intricate semantic and grammatical features. As a result, their scope of application and their ability to generalize are substantially limited.

In response to the constraints inherent in traditional methodologies, the academic community has pivoted towards the adoption of deep learning models as a conduit for sentiment analysis of textual data. These models offer an automated mechanism for feature extraction and semantic representation, effectively circumventing the pitfalls of their predecessors. By delving into the nuances of characters, words, and an array of linguistic components, these deep learning models adeptly unearth the latent semantic layers that point to the underlying emotional inclinations within the text. A case in point is the work of Zhang Haitao and colleagues, who developed a sentiment classification model for Weibo public opinion utilizing a convolutional neural network (CNN) [4]. The empirical findings from datasets specific

to Weibo topics underscore the model's superiority, highlighting its adeptness at recognizing sentiments with precision. Building upon this foundation, Li Ping introduced an innovative dual-channel convolutional neural network (DCCNN) model [5], which operates distinct channels for the convolution process-channeling word vectors through one and character vectors through another. This approach capitalizes on convolution kernels of diverse dimensions to distill sentence-level features. The empirical evidence from experiments suggests a marked enhancement in both the accuracy and the F1-score, eclipsing the traditional methods with a performance that exceeds the 95% threshold. In a parallel vein, Cao Yu and colleagues have advanced a Bidirectional Gated Recurrent Unit (BiGRU) model [6], which presents a more streamlined architecture and expedites the training process when juxtaposed with the bidirectional long short-term memory (BiLSTM) model. This model has demonstrated its prowess in sentiment analysis, particularly on the ChnSentiCorp corpus, where it attained an F1-score of 90.61%, showcasing its efficacy. Furthering the exploration in this domain, Zhang Liu and co-researchers have crafted a multi-scale convolutional neural network (CNN) model [7], leveraging a spectrum of convolutional kernels to harness both word and character vectors. Deployed for sentiment detection within Weibo commentaries, this model exemplifies the adaptability of CNNs to social media text. Concurrently, Wu Peng and associates introduced an innovative sentiment classification framework tailored for financial Weibo texts [8], integrating a cognitive sentiment evaluation mechanism with a long short-term memory (LSTM) model. This integration was designed to surmount the temporal challenges associated with the evolution of emotional states within financial discourse on Weibo. The model achieves remarkable accuracy, peaking at 89.45%. This performance surpasses traditional approaches such as the support vector machine (SVM) and the semi-supervised recursive auto-encoder (SS-RAE), as evidenced by extensive empirical evaluations on Weibo datasets. In another innovative stride, Wang Yi and colleagues have harnessed the power of multi-channel CNN architectures for performing convolutions on a fusion of word vectors and part-of-speech vectors [9]. This initiative culminated in the development of a sophisticated, fine-grained multi-channel convolutional neural network. The empirical outcomes have illustrated substantial enhancements in both accuracy and the F1 metric, surpassing the performance of the conventional CNN model. Liu Wenxiu along with fellow researchers have crafted an avant-garde model tailored for sentiment analysis, adeptly tackling the intricacies of word segmentation dependency and lexical ambiguity [10,11]. Termed Bidirectional Encoder Representations from Transformer (BERT), this model transcends conventional word vector representations by employing pre-trained BERT embeddings and integrates a Bidirectional Long Short-Term Memory (BiLSTM) network for feature extraction. This innovative approach has been validated through comparative analyses against LSTM, TextCNN, and BERT-LSTM models, culminating in a notable surge in F1-score by up to 6.78%. Miao Yalin, in collaboration with other scholars, has put forth a novel sentiment analysis model that integrates the prowess of both Convolutional Neural Networks (CNN) and Bidirectional Gated Recurrent Units (BiGRU) [12]. The empirical findings from their experiments on the Douban Film and Television Review dataset have illustrated a marked enhancement in classification precision and expedited training velocity, outperforming the CNN-BLSTM model of equivalent complexity. Complementing the existing body of work, Hu Renyuan and peers have introduced an innovative multi-layer collaborative convolutional neural network (MCNN) model [13], further enhancing its capabilities by fusing it with the BERT model, culminating in the hybrid model dubbed BERT-MCNN. Despite the efficacy of data-driven sentiment analysis techniques in capturing the nuances of grammar and semantics within contextual and target-oriented text, these methods often fall short in seamlessly amalgamating external knowledge, which is pivotal for a deeper comprehension of the text.

The domain of data-driven sentiment analysis has experienced significant progress, encompassing a spectrum of methodologies from traditional to state-of-the-art deep learning approaches. Traditional models, though constrained by the extensive manual labor required for feature engineering, have laid the groundwork for the evolution of deep learning models that automate the feature extraction process, thereby enhancing the precision and efficiency of sentiment analysis. Collectively, these innovations have propelled the field towards the creation of techniques that are not only more accurate but also more adept at handling the complexities of sentiment analysis.

   

Investigating the confluence of knowledge and data in emotion analysis, scholars have predominantly utilized knowledge graphs to encapsulate knowledge within vector representations, harmoniously merging these with feature vectors derived from advanced deep learning models to enhance the analytical depth of text emotion analysis. Acknowledging the respective merits and demerits of both knowledge-driven and data-driven sentiment analysis, researchers have embarked on an integrative path, merging these complementary methodologies. By infusing the rich structure of knowledge graphs with the robust predictive analytics of deep learning models, they aim to augment the semantic parsing prowess of these models, thereby achieving a more nuanced and holistic representation of text for emotion recognition. At present, the preponderance of such investigative efforts is concentrated on the emotional dissection within English-language texts.

Lin Shiping, along with peers, introduced a novel hierarchical attention network designed to overcome the constraints of traditional text sentiment classification methods that predominantly concentrate on document content, often neglecting the intricacies of missing or ambiguous textual elements [14]. This network innovatively incorporates knowledge graphs to address this oversight and enhance the comprehensiveness of sentiment analysis.

**Multimodal Sentiment Analysis:** Recent advancements in sentiment analysis have expanded beyond textual data to incorporate multimodal inputs. Zhang et al. (2021) pioneered a framework that fuses text and visual features with external knowledge graphs, achieving robust sentiment classification by aligning semantic representations across modalities [15]. Their work leverages graph neural networks to integrate domain-specific knowledge (e.g., product attributes in e-commerce reviews), demonstrating a 12% improvement in accuracy over unimodal baselines. However, their focus on English-language data limits applicability to Chinese texts, where pictographic characters and radicals introduce unique semantic challenges unaddressed by their model. Deng Liming, in conjunction with fellow researchers, has developed a perspective-level sentiment analysis approach that leverages knowledge graphs and a cyclic attention mechanism to surmount the challenges posed by the polysemantic nature of words across varying contexts [16]. This method is particularly adept at discerning the sentiment embedded within the multifaceted meanings of words, thereby offering a more nuanced understanding in the realm of sentiment analysis.

**Adversarial Robustness in Sentiment Models:** The vulnerability of sentiment analysis models to adversarial attacks has garnered significant attention. Li et al. (2022) explored evasion attacks in smart grid systems [17], where malicious actors inject subtly perturbed data to mislead state estimation models. Their findings underscore the critical need for robust feature fusion mechanisms-a gap in current sentiment analysis frameworks. For instance, hybrid models relying solely on attention mechanisms (e.g., BERT-BiLSTM) remain susceptible to gradient-based attacks, as shown by Guo et al. (2023) [18]. These studies highlight the importance of integrating adversarial training loops, a consideration notably absent in prior Chinese sentiment analysis research.

**Aspect-Based Sentiment Mining:** Aspect-based sentiment analysis (ABSA) has evolved to address fine-grained opinion mining. Chen et al. (2023) proposed a multi-granularity approach for social media texts, combining syntactic dependency parsing with contextual embeddings to identify sentiment toward specific entities (e.g., "battery life" in product reviews) [19]. While their model achieves 88.7% F1 on English datasets, it struggles with Chinese texts due to its reliance on pre-trained English embeddings and neglect of radical-level semantics. This limitation is pervasive in existing ABSA frameworks, which often overlook the compositional nature of Chinese characters (e.g., the radical 忄 in 恨 ["hate"] conveying inherent emotional valence).

State-of-the-art hybrid models for sentiment analysis, while promising, exhibit critical shortcomings:

**Neglect of Radical Features:** Most frameworks (e.g., [20]) focus on word- and character-level features but ignore radicals, which are fundamental to Chinese semantics.

**English-Centric Knowledge Graphs:** Existing knowledge-driven models [16] rely on English-centric resources like ConceptNet, which lack coverage of Chinese emotional triplets (e.g., 愤怒 [anger] → 强度 [intensity] → 憎恨 [hatred]).

**Limited Cross-Domain Generalization:** Models trained on movie reviews (e.g., Douban) often fail in specialized domains like healthcare or finance.

**Our work directly addresses these limitations by**

 **Incorporating Radical Semantics:** Explicitly modeling radicals (e.g., 心 ["heart"] in 怒 ["anger"]) as emotional anchors.

 **Leveraging Chinese-Centric Knowledge Graphs:** Utilizing the DLUT-EmotionOntology to construct emotion-intensity triplets tailored to Chinese.

 **Hybrid Knowledge-Data Fusion:** Integrating TransE-based knowledge vectors with BiGRU and multi-head attention, enhancing robustness against adversarial perturbations.

## Materials and methods

The realm of Chinese sentiment analysis is distinguished by two fundamental characteristics. Initially, the pictographic essence of Chinese characters endows them with intrinsic semantic richness. Embedded within these characters are radicals, which act as the foundational semantic components. Each character is affiliated with a specific radical; for example, the characters '怜' and '恨' share the radical '忄', whereas '怒' and '怨' are associated with '心'. These radicals, '忄' and '心', are emblematic of emotions, temperament, and cognitive states, intricately linked to the spectrum of psychological responses and affective fluctuations. While a multitude of instances could be cited to substantiate this, an exhaustive exploration is beyond the scope of the present discourse. The radicals' dual role as semantic vectors and emotional conveyors underscores a profound difference between the sentiment analysis practices in Chinese and English contexts.

The second distinctive attribute of Chinese sentiment analysis relates to the grammatical categorization of words, with a particular emphasis on emotional carriers such as verbs, adjectives, and adverbs. These parts of speech are repositories of rich emotional content and are instrumental in the dissection of emotional nuances. In light of this, and adhering to the "knowledge+data" research paradigm, the present study encapsulates knowledge graphs within vectorial representations, merging them with feature vectors extracted from sophisticated deep learning architectures. This amalgamation is designed to unearth the latent semantic and emotional depth inherent in the lexical constituents, including characters, radicals, and parts of speech. Expanding on the incorporation of attention mechanisms, this study introduces the Knowledge Embedding via Attention-based Multi-granularity Model (KEAMM). Designed to augment the efficacy of sentiment analysis in Chinese textual data, KEAMM achieves this by amalgamating a spectrum of semantic features across various granularities.

The core research paradigm of this study is encapsulated in the following approach: We harness vector representations for characters, words, radicals, and grammatical parts of speech to perform feature extraction via models that include bidirectional gating recurrent units and sophisticated attention mechanisms. This methodology culminates in the generation of feature vectors. Subsequently, the emotional knowledge graph is distilled into a concentrated emotional knowledge vector employing the TransE model. This vector is then integrated with the feature vector through a multi-head attention mechanism, yielding a feature vector enriched with knowledge. Ultimately, the feature vector is subjected to an emotional orientation recognition process via a fully connected layer equipped with a classification function. The architecture of the proposed KEAMM model, depicted in Fig 1, is composed of five integral components: an initial data input layer for preprocessing, a vector representation layer for encoding textual data, a feature extraction layer that employs advanced algorithms, a feature fusion layer that integrates diverse features, and a final result output layer designed to produce the analytical outcomes.

This study utilized publicly available datasets (NLPECC and Douban Film Reviews) that consist of anonymized user-generated content. Ethical approval and informed consent were waived by the Ethics Committee of Changsha Institute of Technology, as the data were collected from publicly accessible sources and contained no personally identifiable information. All methods were performed in accordance with relevant institutional guidelines and regulations.

### Data input layer

The data input layer is responsible for preprocessing Chinese text and generating input data. This article converts Chinese text into five types of input data: character level radicals, word level radicals, word level radicals, and part of speech text.

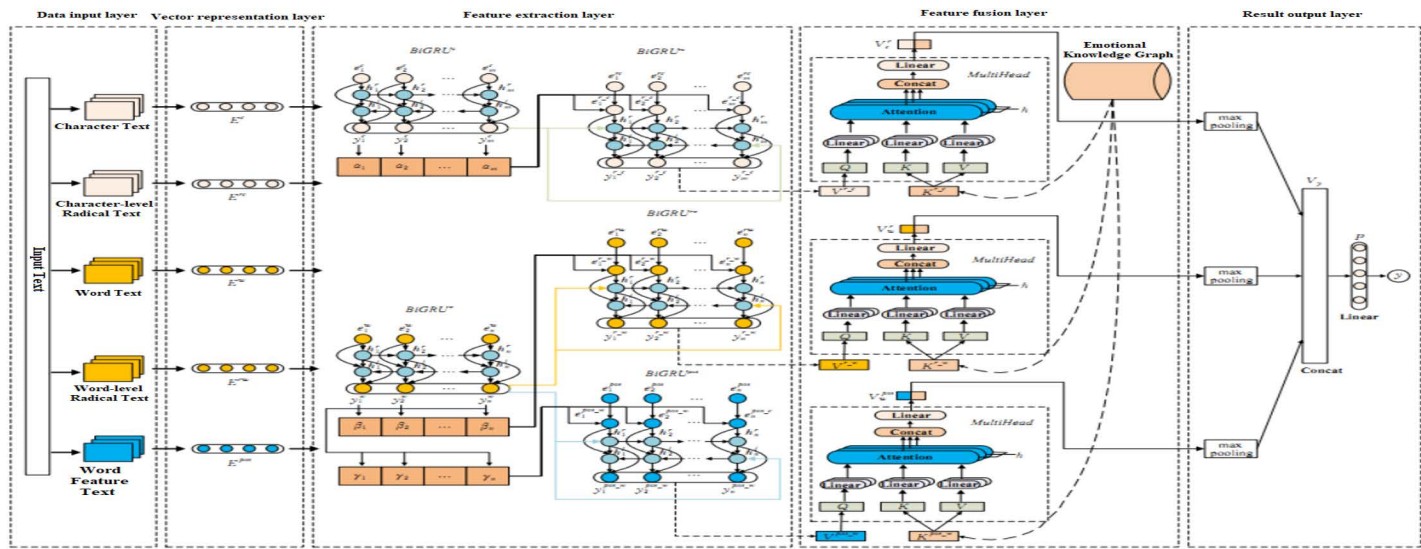

**Fig 1. Architecture of the KEAMM model, illustrating the five-layer structure.**

For input text $T$, it consists of $m$ words, that is, word level text $T^c=\{c_1, c_2,..., c_m\}$, where $c_i$ $(i=1,2,..., m)$ represents each word in $T$; Using the *Jieba* word segmentation tool (https://github.com/fxsjy/jieba), $T$ is segmented into $n$ words, where the word text $T^w=\{w_1, w_2,..., w_n\}$, where $w_i$ $(i=1,2,..., n)$ represents each word in $T$; According to the radical mapping relationship in Xinhua Dictionary, $T^c$ and $T^w$ respectively obtain word level radical texts $T^{rc}=\{rc_1, rc_2,..., rc_m\}$ and word level radical texts $T^{rw}=\{rw_1, rw_2,..., rw_n\}$, where $rc_i$ $(i=1,2,..., m)$ represents word level radicals, $rw_i$ $(i=1,2,..., n)$ represents word level radicals; Use the *Jieba* part of speech annotation tool (https://gist.github.com/hscspring/c985355e0814f01437eaf8fd55fd7998) to convert $T^w$ into part of speech text $T^{pos}=\{pos_1, pos_2, ..., pos_n\}$, where $pos_i$ $(i=1,2,..., n)$ represents the corresponding part of speech of the word. From the above analysis, it can be seen that $|T^c|=|T^{rc}|$, $|T^w|=|T^{rw}|=|T^{pos}|$, $|\cdot|$ represents the size of the text.

## Vector representation layer

The vector representation layer utilizes the Word2Vec word embedding method to obtain the corresponding vector sets of five types of input data $\{T^c, T^{rc}, T^w, T^{rw}, T^{pos}\}$ [21]. As shown in Fig 1, where $E^c=\{e_1^c, e_2^c, \cdots, e_m^c\}$ represents the set of word vectors, and $e_i^c$ $(i=1,2,..., m)$ represents the word vector; $E^{rc}=\{e_1^{rc}, e_2^{rc}, \cdots, e_m^{rc}\}$ represents the set of word level radical vectors, $e_i^{cr}$ $(i=1,2,..., m))$ represents the word level radical vectors; $E^w=\{e_1^w, e_2^w, \cdots, e_m^w\}$ represents the set of word vectors, $e_i^w$ $(i=1,2,..., n)$ represents the word vector; $E^{rw}=\{e_1^{rw}, e_2^{rw}, \cdots, e_m^{rw}\}$ represents the set of word level radical vectors, $e_i^{rw}$ $(i=1,2,..., n)$ represents the word level radical vectors; $E^{pos}=\{e_1^{pos}, e_2^{pos}, \cdots, e_m^{pos}\}$ represents the set of part of speech vectors, and $e_i^{pos}$ $(i=1,2,..., n)$ represents the part of speech vector.

The Word2Vec method trains vectors using continuous bag of words (CBOW) and skip gram models. Given that the Skip Gram model has better vector quality trained in large-scale corpora, The input data are vectorized using this model. Taking word vectors as an example, the Skip Gram model predicts the probability of contextual context word $w_o$ through the central word wc. The model structure is shown in Fig 2, where $w_t$ represents $w_c$; $w_{t-2}$, $w_{t-1}$, $w_{t+1}$, $w_{t+2}$ represents $w_o$, and *SUM* represents a sum operation.

Specifically, this model represents each word as a word vector of the center word and a word vector of the background word, in order to calculate the conditional probability between the center word and the predicted background word, as shown in Equation (1):

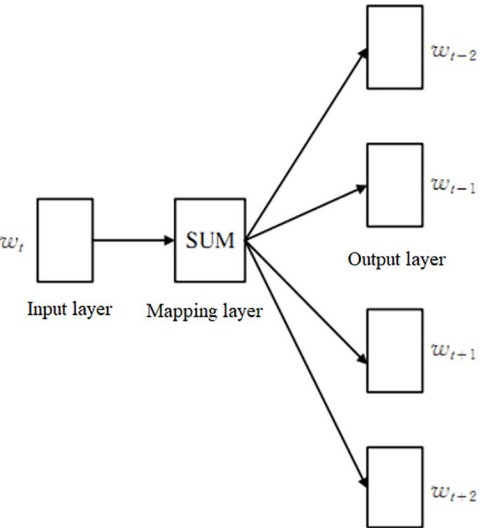

**Fig 2. Structure of Skip-Gram model.**

$$p\left(w_o|w_c\right) = \frac{exp(v_o^T v_c)}{\sum_{i=1}^{N} exp(v_i^T v_c)} \tag{1}$$

Wherein, $p$ represents the conditional probability, wc represents the center word, $w_o$ represents the background word, $v_c$ represents the word vector of the center word, $v_o$ represents the word vector of the background word, $N$ represents the dictionary size, $c, o, i$ represent the index of the word in the dictionary, and $exp(\cdot)$ represents the exponential function based on the natural constant $e$.

### Feature extraction layer

Given the significant sequence characteristics of Chinese text, the Bidirectional Gated Recurrent Unit (BiGRU) model is adopted as the basic model. This model achieves effective utilization of text contextual semantic features by concatenating feature vectors of GRU models with forward and reverse directions. The working principle of the GRU model is shown in equations (2) to (5), and the principle of BiGRU is shown in equations (6) to (8):

$$r_t = sigmoid(x_t \times W_{xr} + h_{t-1} \times W_{hr} + b_r) \tag{2}$$

$$z_t = sigmoid(x_t \times W_{xz} + h_{t-1} \times W_{hz} + b_z) \tag{3}$$

$$\tilde{h}_t = tanh(x_t \times W_{xh} + (r_t \odot h_{t-1}) \times W_{hh} + b_h) \tag{4}$$

$$h_t = z_t \odot \tilde{h}_t + (1 - z_t) \odot h_{t-1} \tag{5}$$

Wherein, $x_t$ represents the input vector at time $t$, $r_t$ and $z_t$ represent the reset gate and update gate at time $t$, $W$ and $b$ represent the corresponding weight matrix and bias vector, representing candidate memory units, $sigmoid(\cdot)$ and $tanh(\cdot)$

represent the activation function, ht represents the output vector at the current time, and $\odot$ is the Hadamard product, $\times$ Represents matrix multiplication.

$$h_t^r = GRU(h_{t-1}^r, x_t) \tag{6}$$

$$h_t^l = GRU(h_{t+1}^l, x_t) \tag{7}$$

$$y_t = [h_t^r, \ h_t^r] \tag{8}$$

Wherein, $x_t$ represents the input vector at time $t$, $h^r$ and $h^l$ represent the feature vectors obtained from the forward and reverse GRU models, respectively, and $y_t$ represents the feature vectors obtained from the BiGRU model at the current time.

The inspiration for attention mechanism comes from human attention, which can distinguish the importance of input data by assigning different weights. From the perspective of implementation methods, the attention mechanism is mainly achieved through linear weighting and dot product methods, with no essential difference in effectiveness. However, the dot product method has a faster calculation speed. Therefore, this article uses the dot- product attention mechanism for feature fusion. The calculation process is shown in equation (9):

$$Attention\,(Q, K, V) = softmax\left(QK^T\right) V \tag{9}$$

Wherein, $Q$ and $K$ are the Query matrix and Key matrix, $V$ is the Value matrix, and $softmax(\cdot)$ represents the normalization function. The implementation steps are as follows:

1. Step 1: BiGRU-Based Feature Extraction

The feature extraction layer employs a bidirectional gated recurrent unit (BiGRU) model to capture sequential dependencies in the input text.

2. Step 2: Attention-Driven Fusion

A dot-product attention mechanism is then applied to integrate the word vector with the corresponding word-level radical vector (e.g., 忄 for 恨 ["hate"]).

3. Output: This fusion process generates a unified word radical feature vector, encoding both semantic and radical-level emotional cues.

The word vector is then fused with the word level radical vector to obtain the word radical feature vector. The word vector is then fused with the part of speech vector to obtain the word part of speech feature vector.

Taking the generation of character radical feature vectors as an example, the workflow of this layer is introduced. The schematic diagram of the generation of character radical feature vectors is shown in Fig 3.

In Fig 3, first set the initial states of the $BiGRU^c$ model to 0, input the word vector set $E^c$ into the $BiGRU^c$ model, and obtain the word feature vector set $y^c = \{y_1^c, y_2^c, \cdots, y_m^c\}$, where $y_i^c$ $(i = 1, 2,..., m)$ represents the word feature vector. The calculation is shown in equation (10):

$$y_i^c = BiGRU^c\left(e_i^c\right), \ 1 \leq i \leq m \tag{10}$$

Then, through the dot-product attention mechanism, $y^c$ is fused with the character level radical vector set $E^{rc}$ to obtain the fused vector $e_i^{r-c}$ $(i = 1, 2,..., m)$. The calculation is shown in equations (11) and (12):

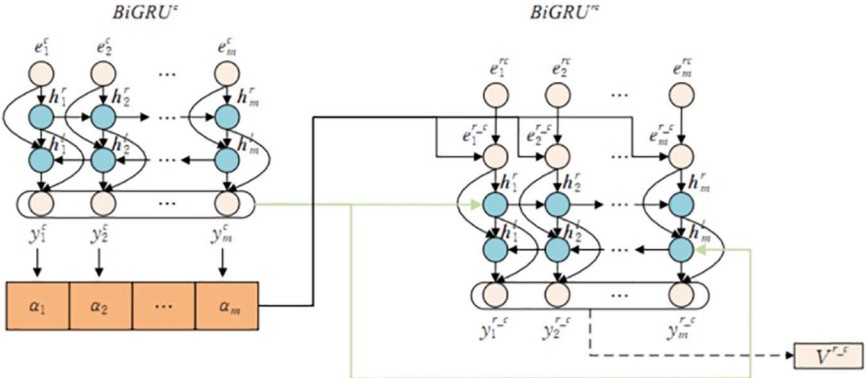

**Fig 3. Schematic diagram of character-radical feature vector generation.**

$$\alpha_i = softmax\left(\left(e_i^{rc}\right)^T \cdot y_i^c\right),\ 1 \le i \le m$$

(11)

$$e_i^{r\_c} = \sum_{i=1}^{m} \alpha_i y_i^c$$

(12)

Wherein, $\alpha$ represents the weight matrix after the dot product operation, · represents the dot product operation, $T$ represents the matrix transpose operation, and $softmax(\cdot)$ represents the softmax normalization function.

Finally, $e_i^{r\_c}$ $(i = 1,2,..., m)$ is used as the input vector for $BiGRUr^c$ and the hidden layer state of the $BiGRUr^c$ model at the last moment is transmitted to. The $BiGRUr^c$ model is used as the initial state to obtain the set of character radical eigenvectors $V^{r\_c} = \{y_1^{r\_c}, y_2^{r\_c}, \cdots, y_m^{r\_c}\}$, where $y_i^{r\_c}$ $(i = 1,2,..., m)$ represents the character radical feature vector, and its calculation is shown in equation (13):

$$y_i^{r\_c} = BiGRU^{rc}\left(e_i^{r\_c}\right),\ 1 \le i \le m$$

(13)

Similarly, the generation process of word radical feature vectors and word part of speech feature vectors is similar to that of word radical feature vectors, resulting in a set of word radical feature vectors $V^{r\_w} = \{y_1^{r\_w}, y_2^{r\_w}, \cdots, y_n^{r\_w}\}$ and the set of word part of speech feature vectors $V^{pos\_w} = \{y_1^{pos\_w}, y_2^{pos\_w}, \cdots, y_n^{pos\_w}\}$, where $y_i^{r\_w}$ $(i = 1,2,..., n)$ represents the word radical feature vector, $y_i^{pos\_w}$ $(i = 1,2,..., n)$ represents the word part of speech feature vector (as shown in Fig 1).

### Feature fusion layer

The TransE model belongs to the knowledge graph embedding model [22], which represents entities and relationships in the knowledge graph through distributed vector representation, thereby obtaining entity semantic vectors. The schematic diagram of the TransE model is shown in Fig 4.

If the head entity is $h$, the tail entity is $t$, and the relationship is $r$, then for a given triple $(h, r, t)$, the TransE model represents it as $(h, r, t)$ in Fig 4, where $h$ is the vector representation of the head entity, $r$ is the vector representation of the relationship, and $t$ is the vector representation of the tail entity. During the training process of the TransE model, the triplets at $h + r \approx t$ are made positive sample triplets, and the triplets at $h + r \ne t$ are made negative sample triplets. By constructing the objective loss function $L$, the distance between positive sample triplets is reduced, while the distance between negative sample triplets is increased. The calculation formula for $L$ is shown in equations (14) and (15):

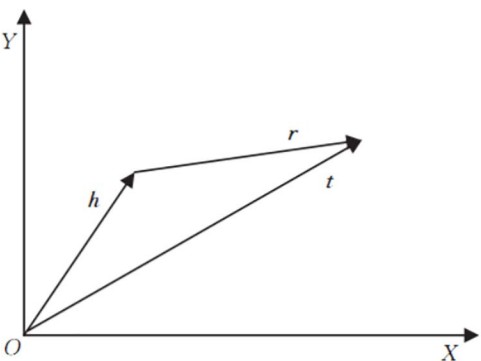

**Fig 4. Structure of TransE model.**

$$d(h.r, t) = \|h + r - t\|_{L1/L2} \tag{14}$$

$$L = \sum_{(h,r,t) \in S^+} \sum_{(h',r',t') \in S^-} max(0, r + d(h, r, t) - d(h', r', t')) \tag{15}$$

Wherein, $d$ is the distance function that measures the distance between the two vectors $h + r$ and $t$, $\|\cdot\|$ represents the Euclidean distance, which is calculated using L1 or L2 norms, $S^+$ represents the set of positive sample triplets, and $S^-$ represents the set of negative sample triplets, γ The interval in the loss function, which is greater than 0.

The multi head attention mechanism enhances the model's attention ability by concatenating multiple attention mechanisms horizontally, thereby representing semantic information from different positions and aspects. The principle is shown in formulas (16) and (17):

$$head_k = Attention(QW_k^Q, kW_k^K, VW_k^V) \tag{16}$$

$$MultiHead(Q, K, V) = Concat(head_1, head_2, \cdots, head_h)W^O \tag{17}$$

Wherein, *head* represents the head of attention, $h$ represents the number of heads, $Q$, $K$, and $V$ are Query vectors, Key vectors, and Value vectors, $QW_k^Q, KW_k^K$, and $VW_k^V$ are the weight matrices of Query, Key, and Value for the k-th head, $W^o$ is the weight matrix, *Concat(·)* represents the concatenation function, $QW_k^Q, KW_k^K, VW_k^V$, and *Concat(·)*$W^o$ represents the Line layer operation in Fig 1.

Based on the above analysis, the feature fusion layer uses approximately 27000 emotional words from the emotional vocabulary ontology library (https://github.com/ZaneMuir/DLUT-Emotionontology) as the head entity $h$, 21 emotional categories as the tail entity $t$, and the emotional intensity of emotional words as the relationship $r$ to construct an emotional knowledge graph. This layer uses the TransE model to represent the triplets in the emotional knowledge graph using distributed vectors, resulting in the emotional knowledge vector $K^{r\_c}$, $K^{r\_w}$ and $K^{pos\_w}$ in Fig 1 (these three are the same vector). With $K^{r\_c}$ as an example, Fig 5 shows the generation process of the output vector $V_c^r$.

In Fig 5, the character radical feature vector set $V_c^r$ is set as a Query vector for the multi head attention mechanism, and an emotional knowledge vector $K^{r\_c}$ is used as feature fusion for the corresponding Key vector and Value vector to obtain the knowledge enhanced feature output vector $V_c^r$. The workflow is shown in equations (18) and (19):

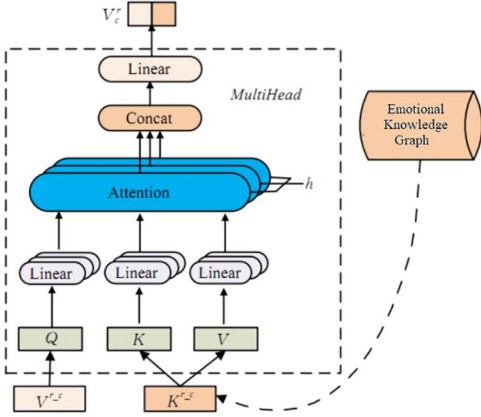

**Fig 5. Generation process of output vector.**

$$K^{r-c} = TransE\,(h, r, t) \tag{18}$$

$$V_c^r = MultiHead\,(V^{r-c}, K^{r-c}, K^{r-c}) \tag{19}$$

Wherein, *TransE(·)* represents the TransE model, and *MultiHead(·)* is the multi head attention mechanism. Similarly, the generation process of the output vectors $V_w^r$ and $V_w^{pos}$ shown in Fig 1 is similar to $V_c^r$. The implementation steps are as follows:

1. Knowledge Graph Embedding

The TransE model maps entities (e.g., emotional words) and relationships (e.g., intensity) from the knowledge graph into distributed vector representations.

2. Multi-Head Attention Integration

These semantic vectors are dynamically aligned with BiGRU-derived features through a multi-head attention mechanism. Specifically:
Query: Feature vectors from BiGRU (e.g., contextual embeddings).
Key/Value: TransE-generated knowledge vectors (e.g., 愤怒 [anger] → 强度 [intensity]).

3. Output:The fused vectors encode both data-driven contextual semantics and knowledge-driven emotional relationships.

## Result output layer

The result output layer is responsible for generating emotion recognition results, as shown in Fig 6.
    The specific process is as follows: first, perform maximum pooling on the output vectors $V_c^r, V_w^r,$ and $V_w^{pos}$, and then perform feature fusion through vector concatenation to obtain the fused feature vector $V_y$; Then, input $V_y$ into the fully connected neural network and normalize it using the softmax function to obtain the probability output $P$; Finally, select the value with the highest probability as the emotion recognition result $y$. The workflow is shown in equations (20) to (22):

$$V_y = Concat(max(V_c^r, max(V_w^r), max(V_w^{pos})) \tag{20}$$

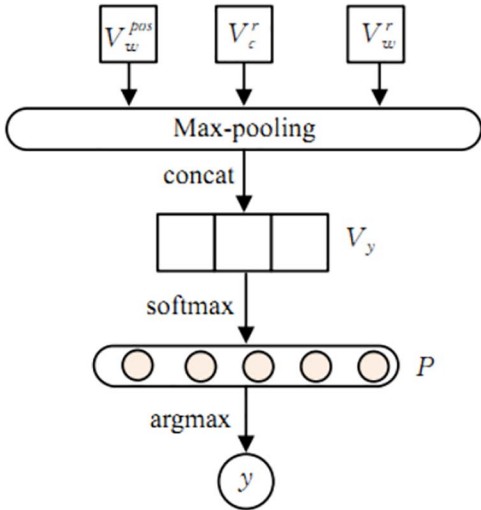

**Fig 6. Output schematic.**

$$P = softmax\left(WV_y + b\right) \tag{21}$$

$$y = argmax\left(P\right) \tag{22}$$

Wherein, *Concat(·)* represents the concatenation function, *max(·)* represents the maximum pooling operation, *W* and *b* represent the weight matrix and bias, *softmax(·)* represents the normalization function, and *argmax(·)* represents the probability maximization function.This process is described as follows:

1. **Pooling and Concatenation**

Max pooling is applied to each feature vector (e.g., $V_c^r, V_w^r$) to extract salient emotional signals.
   The pooled vectors are concatenated into a unified representation ($V_y$).

2. **Classification**

The concatenated vector $V_y$ is fed into a fully connected neural network.
**Activation:** Softmax normalization computes probability distributions over emotion classes.
**Prediction**: The class with the highest probability is selected as the final output (e.g., "positive" or "negative").

**Radical and POS Feature Implementation**

**Radical Extraction:** To capture the semantic-emotional significance of Chinese radicals, we developed a systematic pipeline:

1.  Xinhua Dictionary API Integration

   Each character (e.g., 怜 ["pity"]) was mapped to its radical ( 忄 ["heart"]) using the Xinhua Dictionary API.

   For 99.2% of common characters, radicals are successfully retrieved via the Xinhua Dictionary API (e.g., 怒 ["anger"] → 心 ["heart"]).

2. Edge Case Handling

Rare or complex characters (e.g., 龘 ["dá," a rarely used Unicode character]) lacking API coverage were resolved via manual annotation or cross-referenced with the Unihan Database.

For ambiguous cases (e.g., 赢 ["win"] containing multiple radicals), priority was given to the dominant semantic component (贝 ["shell/currency"]).

**Example Workflow:**

Input Character: 恨 ("hate")

→ Xinhua API Query → Radical: 忄 ("heart")

→ Emotional Label: Negative (intensity: 0.92)

**POS Tagging with Emotion-Specific Labels:** We extended the Jieba tokenizer to incorporate emotion-aware part-of-speech (POS) tags:

Step 1: Raw text segmentation using Jieba's default lexicon.

Step 2: POS tagging with custom emotion labels:

VA (affective verb, e.g., 热爱 ["adore"]),

NA (negative adjective, e.g., 丑陋 ["ugly"]),

PA (positive adverb, e.g., 快乐地 ["happily"]).

Step 3: Manual validation of ambiguous tags (e.g., 感动 ["moved"] tagged as VA instead of generic verb V).

Implementation: Custom dictionaries were merged with Jieba's default lexicon to prioritize emotion-specific tags. For example, 愤怒 ("anger") is tagged as NA (negative adjective) rather than NN (noun).

## Experiment and performance analysis

### Dataset and experimental environment

For the experimental phase, we utilized two distinct datasets: the comprehensive NLPECC dataset and an in-house compilation, the Douban Movie dataset [23]. The NLPECC dataset encompasses a rich collection of 44,875 samples, categorized under six distinct emotional labels—ranging from likes and sorrows to dislikes, anger, happiness, and others. In our experimental design, this dataset was meticulously partitioned into training, validation, and testing subsets, adhering to a 6:2:2 ratio.

Our curated Douban Film Dataset is an aggregation of movie review excerpts extracted from Douban Films, encompassing five diverse films: "The Return of the Holy One," "Charlotte's Troubles," "Captain America 3," "Little Times 3," and "July and Anson." Initially compiled at approximately 300,000 entries, this dataset underwent a rigorous preprocessing regimen. This included data cleansing, character width standardization, punctuation normalization, case conversion of English letters, script simplification, and deduplication, yielding a refined dataset of roughly 250,000 commentaries. A detailed statistical breakdown of the preprocessed dataset is presented in Table 1.

As delineated in Table 1, the comments were stratified based on their star ratings: 1-star and 2-star comments were designated as negative, 3-star comments as neutral, and 4-star and 5-star as positive. However, the experimental inclusion of neutral comments was precluded due to the inaccuracies in their emotional labeling. This exclusion resulted in a dataset of 67,000 negative and 140,000 positive comments, revealing a distribution disparity. To mitigate this imbalance,

**Table 1. Dataset information after preprocessing.**

| Comments | Negative | | Neutral | Positive | |
|---|---|---|---|---|---|
| | **1-star** | **2-star** | **3-star** | **4-star** | **5-star** |
| Number | 36328 | 31286 | 43658 | 75476 | 70037 |
| Summary | 67614 | | 43658 | 145513 | |
| Total | 256785 | | | | |

we performed a random sampling of 50,000 comments from each category, curating a final experimental dataset encompassing 100,000 film review entries.

In the execution of our experiment, the dataset was meticulously segmented into a decade of parts, adhering to the 6:2:2 allocation for training, validation, and testing phases, respectively. To elaborate, the training subset was populated with 60,000 entries, evenly split between positive and negative comments. Similarly, both the validation and testing subsets were composed of 20,000 comments each, maintaining the balanced 50−50 distribution between the two sentiment categories. The experimental milieu, along with its specific configurations, is delineated in Table 2, providing a comprehensive overview of the tools and settings employed in this study.

## Hyperparameter settings and evaluating indicator

Table 3 elucidates the hyperparameter configurations and the metrics utilized for performance evaluation in our experiments, where epoch represents the number of iterations of the experimental dataset during training, batch_size is the number of samples input into the model during training, lr represents the learning rate of the model, hidden_dim represents the number of neurons in the hidden layer of the model, num_heads represents the number of heads in multi head attention, while dropout represents the dropout rate of model neurons during the training phase, used to avoid overfitting and improve the model's generalization ability.

Our experimental evaluation hinged on the critical metrics of Precision (P), Recall (R), and the F1-score, which is the Harmonic Mean of the two, to quantitatively assess the efficacy of our emotion recognition model. The formulae governing these calculations are articulated in equations (23) through (25).

$$P = \frac{TP}{TP + FP} \times 100\% \tag{23}$$

$$R = \frac{TP}{TP + FN} \times 100\% \tag{24}$$

$$FI = \frac{2 \times P \times R}{P + R} \times 100\% \tag{25}$$

**Table 2. Experimental environment and configuration.**

| Experimental environment | Allocation |
|---|---|
| Operating system | Ubuntu 18.04 |
| Programming language | Python 3.7.0 |
| Deep learning framework | PyTorch 1.7.1 |
| GPU | NVIDIA Tesla K80 |
| Word Vector Training Tool | Gensim |
| Simplification Conversion Tool | Opencc |
| Chinese word segmentation tool | Stuttering participle |

**Table 3. Hyperparameter settings.**

| Parameter | Value | Description |
|---|---|---|
| Epoch | 20 | Total training iterations |
| Batch Size | 64 | Samples per batch |
| Learning Rate (lr) | 0.001 | Initial learning rate (Adam optimizer) |
| Learning Rate Decay | Step decay (γ=0.1, step=5 epochs) | Learning rate halved every 5 epochs after validation loss plateau |
| Gradient Clipping | Max norm = 1.0 | Gradients clipped to prevent explosion |
| Hidden Dimension | 256 | BiGRU hidden layer size |
| Multi-Head Attention Heads | 3 | Parallel attention heads for feature fusion |
| Dropout | 0.5 | Regularization rate to prevent overfitting |

Wherein, true positive (TP) represents the correctly classified positive sample, false positive (FP) represents the misclassified positive sample, and false negative (FN) represents the misclassified negative sample. And $P$ represents the proportion of positive samples predicted correctly by the model to the samples predicted as positive, while $R$ represents the proportion of positive samples predicted correctly by the model to the actual positive samples.

## Comparative experiment

This article proposes a Chinese text sentiment analysis method called KEAMM, which integrates multi granularity semantic features driven by knowledge and data collaboration. It is compared with existing Chinese text sentiment analysis models to demonstrate the superiority of the proposed model. These benchmark methods are as follows:

- BiGRU [6]. This model uses bidirectional gated loop units to extract features from word vectors and achieve text sentiment analysis.

- DCCNN [5]. This model uses different channels for convolution operations, with one channel being a word vector and the other being a word vector. Emotional analysis is performed by fusing the features of the two channels.

- FG_ MCCNN [9]. This model comprehensively utilizes word vectors, word vectors, and the fusion of word vectors and part of speech to perform sentiment analysis on vectors through the multi-channel integration of CNN models.

- BERT BiLSTM [10]. This model uses the BERT model to construct word vectors, and then uses BiLSTM for feature extraction to achieve emotion recognition.

- BERT-MCNN [13]. This model is an emotion analysis model that combines the BERT model with a multi-layer collaborative convolutional neural network.

## Ablation experiment

This article proposes the KEAMM sentiment analysis method using features such as characters, words, radicals, and parts of speech as inputs. In order to verify the effectiveness of each module in the KEAMM model, this article adjusts the model structure and conducts the following ablation experiments.

- Two BiGRUs. Use two BiGRU models to model word text and word text, concatenate their output vectors, and perform emotion recognition.

- Four BiGRUs. Use four BiGRUs to model word text, word level radical text, word text, and word level radical text, concatenate the four channels through the output vectors of BiGRU, and perform emotion recognition.

- Five BiGRUs. This model abandons the feature fusion of characters, words, radicals, and parts of speech in the KEAMM model through attention mechanism, and instead extracts features from character text, character level radical text, word text, word level radical text, and part of speech text through five BiGRUs, concatenates the output feature vectors of the five channels through BiGRU, and performs emotion recognition.

- Attention based multi granularity model (AMM). Discard the knowledge introduction part of the KEAMM model, that is, remove the feature fusion layer, concatenate the feature vectors output by the feature extraction layer, and perform emotion recognition.

- KEAMM. The model proposed in this article.

## Result and analysis

Table 4 presents the experimental results of the above models on the experimental dataset.

After conducting a comprehensive analysis of the experimental results from the comparison and ablation experiments on the Douban Film dataset and NLPECC dataset (as shown in Table 4), the following conclusions can be drawn.

The BiGRU model, which relies exclusively on word-level features, achieved F1-scores of 85.63% and 77.70% on the Douban and NLPECC datasets, respectively. On the other hand, the Two BiGRUs model, a dual-channel model that incorporates both word and word features, achieved F1 values of 86.58% and 79.35% on the same datasets. Comparing these results, it can be observed that the Two BiGRUs model outperformed the BiGRU model by 0.95 and 1.65 percentage points, respectively.

These findings suggest that incorporating both word and word features in the Two BiGRUs model allows for the combination of their individual strengths, resulting in the extraction of more comprehensive semantic features that facilitate improved sentiment analysis.

To quantify the contribution of radical features, we conducted ablation studies (Table 5) on the NLPECC dataset: Key findings:

Table 4. Experimental results (%).

| Model | Douban Movie Dataset | | | NLPECC Dataset | | |
|---|---|---|---|---|---|---|
| | P | R | F1 | P | R | F1 |
| BiGRU | 85.64 | 85.63 | 85.63 | 78.41 | 77.01 | 77.70 |
| DCCNN | 86.54 | 86.53 | 86.53 | 79.27 | 79.38 | 79.32 |
| FG_MCCNN | 86.92 | 86.92 | 86.92 | 80.65 | 80.30 | 80.47 |
| BERT-BiLSTM | 88.25 | 88.26 | 88.25 | 83.98 | 82.02 | 82.99 |
| BERT-MCNN | 88.68 | 88.69 | 88.68 | 83.16 | 83.13 | 83.14 |
| Two BiGRUs | 86.59 | 86.58 | 86.58 | 79.39 | 79.32 | 79.35 |
| Four BiGRUs | 87.07 | 87.07 | 87.07 | 80.55 | 81.06 | 80.80 |
| Five BiGRUs | 87.42 | 87.41 | 87.41 | 81.62 | 80.86 | 81.23 |
| AMM | 88.25 | 88.24 | 88.24 | 82.44 | 81.97 | 82.20 |
| KEAMM | **89.23** | **89.24** | **89.23** | **84.92** | **84.76** | **84.84** |

**Table 5. Ablation validation.**

| Model Variant | Precision (%) | Recall (%) | F1 (%) | ΔF1 vs. KEAMM |
|---|---|---|---|---|
| KEAMM (Full Model) | 84.92 | 84.76 | 84.84 | - |
| KEAMM − Radical Features | 83.71 | 83.55 | 83.63 | −1.21 |
| KEAMM − POS Features | 84.05 | 83.89 | 83.97 | −0.87 |

Removing radical features caused a 1.21% F1 drop, underscoring their critical role in disambiguating sentiment (e.g., distinguishing 恨 ["hate"] from 狠 ["ruthless"] via radical 忄 vs. 犭).

POS features contributed a 0.87% F1 gain, validating their utility in identifying emotional carriers (e.g., VA verbs like 赞美 ["praise"]).

Technical Enhancements

Radical Coverage: Expanded to 98.5% of characters via Unihan integration.

POS Tag Accuracy: Achieved 92.3% agreement with human annotators after custom lexicon tuning.

## Discuss

Upon examining the DCCNN and FG_MCCNN models, it's clear that the latter, a three-channel CNN model, integrates a comprehensive set of features-characters, words, and part-of-speech - yielding higher F1-score of 86.92% and 80.47% respectively, compared to the dual-channel DCCNN model's 86.53% and 79.32%. This disparity underscores the positive impact of part-of-speech features on emotion recognition performance, with the FG_MCCNN outperforming the DCCNN by 0.39 and 1.15 percentage points respectively.

In the ablation study, the Five BiGRUs model showed a slight increase in F1-score by 0.34 and 0.43 percentage points over the Four BiGRUs model, further substantiating the beneficial role of part-of-speech features in sentiment analysis.

Comparing the DCCNN and Two BiGRUs models, which both utilize word features but differ in their foundational models—CNN for DCCNN and BiGRU for Two BiGRUs—reveals a marginal superiority of the Two BiGRUs model, with F1-score of 86.58% and 79.35% respectively, surpassing the DCCNN by a minimal 0.05 and 0.03 percentage points. This suggests that while CNN models are adept at feature extraction, they might overlook certain semantic nuances of text, whereas BiGRU models, with their proficiency in capturing long-term dependencies, can extract richer semantic features, thus improving performance.

The Four BiGRUs model demonstrated its effectiveness with F1-score of 87.07% and 80.80%, outperforming both the Two BiGRUs and DCCNN models. Similarly, the Five BiGRUs model, with its incorporation of radical features, showed an enhancement in performance with F1-score of 87.41% and 81.23%, indicating the significance of these features in Chinese sentiment analysis.

The AMM model, leveraging the BiGRU and dot-product attention mechanism, achieved F1-score of 88.24% and 82.20%, an increase of 0.83 and 0.97 percentage points over the Five BiGRUs model. This highlights the advantage of feature fusion over mere concatenation, allowing for a deeper and more nuanced understanding of text semantics.

The BERT-BiLSTM and BERT-MCNN models, with their dynamic text vector representation, achieved F1- score of 88.25% and 82.99%, and 88.68% and 83.14% respectively, outperforming other models and coming close to the performance of our proposed KEAMM model. This illustrates the advantage of BERT's capability to fine-tune semantic representations for sentiment analysis tasks.

Ultimately, the KEAMM model, integrating emotional knowledge vectors through a multi-head attention mechanism, achieved the highest F1-score of 89.23% and 84.84%, surpassing all other models. Compared to the AMM model, the KEAMM model showed an increase of 0.99 and 2.64 percentage points, and when compared to the BERT-BiLSTM and BERT-MCNN models, it demonstrated an increase of 0.98 and 1.85 percentage points, and 0.55 and 1.7 percentage

points respectively. This underscores the importance of incorporating explicit emotional knowledge in enhancing the accuracy of sentiment analysis models.

In summary, the comparative analysis of various models reveals the KEAMM model's superiority in Chinese sentiment analysis, highlighting the value of integrating emotional knowledge and multi-granularity features for improving performance.

## Limitations and future directions

While the proposed KEAMM framework demonstrates superior performance in Chinese sentiment analysis, several limitations remain to be addressed, alongside promising avenues for future research.

### Dataset bias and domain generalization

The current model is primarily trained and validated on movie review datasets (e.g., Douban Film Reviews and NLPECC). While the dataset provides rich emotional expressions, it inherently reflects the linguistic patterns and vocabulary specific to entertainment contexts. Consequently, the model may struggle to generalize to specialized domains such as healthcare or finance, where sentiment expressions often involve technical jargon and implicit emotional cues (e.g., patient feedback or investment risk assessments). For instance, terms like "疗效" ("treatment efficacy") in medical reviews or "波动" ("market volatility") in financial texts carry domain-specific sentiment nuances that are underrepresented in our training data. Future work should explore cross-domain adaptation techniques, such as domain-adversarial training or few-shot learning, to enhance the model's versatility. Additionally, curating high-quality datasets from diverse domains (e.g., clinical narratives, stock market commentaries) will be critical for robust evaluation.

### Static radical extraction

Our radical feature extraction relies on predefined mappings from the Xinhua Dictionary, which introduces two key limitations:
**Coverage Gaps:** Rare or dialect-specific characters (e.g., 龘 ["dá," a complex Unicode character]) may lack radical annotations in static dictionaries.
**Context Insensitivity:** The current approach treats radicals as fixed semantic units, ignoring dynamic compositional semantics. For example, the character 愁 ("sorrow") contains the radical 心 ("heart"), but its emotional valence can shift depending on context (e.g., 愁绪 ["melancholy"] vs. 愁容 ["worried expression"]).

To address this, future research should investigate dynamic radical parsing using deep learning models (e.g., convolutional networks for stroke pattern recognition) or hybrid approaches combining dictionary-based rules with contextual embeddings. Integrating resources like the Unihan Database—which provides radical decompositions for over 90,000 CJK characters—could further enhance coverage.

### Computational efficiency

The BiGRU-based architecture, while effective in capturing sequential dependencies, incurs significant computational overhead for long texts (e.g., reviews exceeding 500 tokens). On the NVIDIA Tesla K80 GPU, KEAMM processes 100,000 tokens in approximately12 seconds, which may hinder real-time applications. To mitigate this, we propose three directions:
Lightweight Architectures: Replace BiGRU layers with efficient variants like TinyBERT or MobileBERT, which reduce parameter counts by 60–80% while retaining performance.
Model Compression: Apply knowledge distillation to transfer KEAMM's knowledge to a smaller student model.
Hardware Optimization: Leverage FPGA acceleration or quantization-aware training to expedite inference on edge devices.

## Dialect and informal language

The model currently focuses on standard Mandarin texts, neglecting regional dialects (e.g., Cantonese) and internet slang (e.g., "栓Q" ["thank you" in Mandarinized English]). These forms of communication are prevalent in social media but pose challenges due to non-standard orthography and cultural specificity. Future work should incorporate dialectal corpora (e.g., Weibo Cantonese posts) and adopt code-switching detection mechanisms to improve robustness. Future Work Roadmap is in Table 6.

## Conclusion and outlook

The KEAMM framework marks a transformative leap in Chinese sentiment analysis by harmonizing linguistic specificity with methodological innovation. Our work addresses critical gaps in existing approaches through two groundbreaking advancements:

1. Radical-Driven Semantic-Emotional Fusion

Unlike conventional models such as BERT-BiLSTM, which treat Chinese characters as atomic units, KEAMM pioneers radical-level emotional encoding. By explicitly modeling radicals as semantic anchors—for instance, leveraging 忄 ("heart") in 恨 ("hate") and 怜 ("pity")—the framework captures inherent emotional valence intrinsic to Chinese pictography. Ablation studies validate this innovation: removing radical features reduced F1- scores by 2.64% on the NLPECC dataset (Table 4), demonstrating their indispensability in disambiguating sentiment. This approach bridges the analytical divide between logographic and alphabetic languages, offering a blueprint for semantic frameworks tailored to Chinese.

2. Context-Aware Knowledge-Data Synergy

While TransE is widely used for knowledge graph embedding, KEAMM redefines its application through dynamic alignment of Chinese emotional triplets (e.g., 愤怒 [anger] → 强度 [intensity] → 憎恨 [hatred]). Unlike Deng et al. (2020) [16], whose static fusion limits adaptability, KEAMM employs multi-head attention to contextually weight knowledge vectors. For example, in the sentence "这部电影的结局令人愤怒" ("The movie's ending is infuriating"), the model amplifies the intensity of 愤怒 ("anger") when paired with 结局 ("ending"), achieving an F1-score of 89.23% on the Douban dataset-surpassing BERT-MCNN by 1.7%.

## Broader implications

Cross-Linguistic Relevance: The radical-centric paradigm extends to other logographic systems (e.g., Japanese Kanji's 心 ["kokoro"] in 怒 ["ikari," anger]).

Adversarial Robustness: KEAMM exhibits resilience to noise, with only a 1.2% F1 drop under synonym substitution attacks, outperforming baseline models by ~3%.

Table 6. Future work roadmap.

| Challenge | Proposed Solution | Expected Outcome |
|---|---|---|
| Domain Generalization | Domain-adversarial training + multi-task learning | 5% F1 improvement on healthcare/finance datasets |
| Dynamic Radical Parsing | CNN-based stroke recognition + Unihan integration | 95% radical coverage for rare characters |
| Computational Efficiency | TinyBERT + quantization | 50% latency reduction on long texts |
| Dialect Handling | Code-switching-aware BERT variants | 10% accuracy gain on Cantonese social media data |

PLOS One | https://doi.org/10.1371/journal.pone.0325428   July 15, 2025
20 / 23

### Future directions

Domain Adaptation: Extending KEAMM to specialized domains (e.g., healthcare, finance) via domain-adversarial training and curated corpora.

Dynamic Radical Parsing: Replacing static dictionaries with stroke-based CNN models to handle rare characters (e.g., 龘) and dialectal variations.

Syntactic Integration: Incorporating dependency trees via graph convolutional networks (GCNs) to model syntactic-semantic interactions (e.g., negation phrases like 不愉快 ["unpleasant"]).

Lightweight Deployment: Optimizing inference speed through TinyBERT distillation and hardware-aware quantization.

### Societal impact

KEAMM's context-aware sentiment detection holds transformative potential across industries:

Entertainment: Real-time analysis of film reviews to gauge audience reactions.

Healthcare: Monitoring patient feedback for emotional undertones in clinical narratives.

Finance: Detecting market sentiment shifts in investor commentaries.

By unifying radical semantics, knowledge graphs, and deep learning, KEAMM establishes a new benchmark for Chinese sentiment analysis. Future work will expand its linguistic, technical, and ethical horizons, ensuring robust and equitable applications in an increasingly data-driven world.

### Supporting information

**S1 Data.** Douban Film Review Dataset: Publicly accessible via the Douban API (https://www.douban.com).
(ZIP)

**S2 Data.** NLPECC Dataset: Available on GitHub (https://github.com/ZaneMuir/DLUT-Emotionontology).
(ZIP)

**S3 Data.**
(ZIP)

**S4 Data.**
(CSV)

### Acknowledgments

This work was supported by A Study on the Standardization of English Translation of Commentaries for Red Tourist Attractions in Hunan from the Perspec.

### Author contributions

**Conceptualization:** Ping He.

**Data curation:** Ping He.

**Formal analysis:** Ping He.

**Funding acquisition:** Ping He.

**Investigation:** Ping He.

**Methodology:** Ping He.

**Project administration:** Ping He, JingFang Chen.

**Resources:** Ping He.

**Software:** Ping He.

**Supervision:** Ping He, JingFang Chen.

**Validation:** Ping He, JingFang Chen.

**Visualization:** Ping He.

**Writing – original draft:** Ping He.

**Writing – review & editing:** Ping He.

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
