## [Decision Letter · Decision Letter 0]

Dear Dr. He,

Thank you for submitting your manuscript to PLOS ONE. After careful consideration, we feel that it has merit but does not fully meet PLOS ONE’s publication criteria as it currently stands. Therefore, we invite you to submit a revised version of the manuscript that addresses the points raised during the review process.

We look forward to receiving your revised manuscript.

Kind regards,

Jiwei Tian

Academic Editor

PLOS ONE

Journal Requirements:

This work was supported by A Study on the Standardization of English Translation of Commentaries for Red Tourist Attractions in Hunan from the Perspective of Communication Theory of Translation (22WLH29) & Research on the Practice of Ideological and Political Education in Business English Courses in Local Undergraduate Colleges - Taking "English for International Business Negotiation as an Example" (HNJG-2022-1157).

6. We note that your Data Availability Statement is currently as follows: All relevant data are within the manuscript and its Supporting Information files.

7. Please upload a new copy of Figure 1, 2, 3, 4, 5, 6 as the detail is not clear. Please follow the link for more information: https://blogs.plos.org/plos/2019/06/looking-good-tips-for-creating-your-plos-figures-graphics/
https://blogs.plos.org/plos/2019/06/looking-good-tips-for-creating-your-plos-figures-graphics/

Reviewers' comments:

Reviewer's Responses to Questions

**Comments to the Author**

1. Is the manuscript technically sound, and do the data support the conclusions?

Reviewer #1: Yes

Reviewer #2: No

2. Has the statistical analysis been performed appropriately and rigorously?

Reviewer #1: Yes

Reviewer #2: No

3. Have the authors made all data underlying the findings in their manuscript fully available?

Reviewer #1: No

Reviewer #2: Yes

4. Is the manuscript presented in an intelligible fashion and written in standard English?

Reviewer #1: Yes

Reviewer #2: No

Reviewer #1: The paper proposes a method that deeply fuses the knowledge vector obtained from the emotional knowledge triplet using the TransE model with feature vectors from models like bidirectional gating loop units and attention mechanisms. It's novel. To further improve the manuscript, the following suggestions are given:

1、In the paper, some figures in the manuscript are a little blurry, please improve the clarity.

2、Since there are some papers in this topic, the contributions of the manuscript should be better summarized and listed.

3、While the introduction sets the context, a more explicit literature review section could better situate the study within the broader research landscape, such as Multimodal Aspect-Based Sentiment Analysis with External Knowledge and Multi-granularity Image-Text Features, EVADE Targeted Adversarial False Data Injection Attacks for State Estimation in Smart Grid, Aspect-based sentiment analysis with multi-granularity information mining and sentiment hint, LESSON Multi-Label Adversarial False Data Injection Attack for Deep Learning Locational Detection, and so on. These references could provide valuable insights into your research.

4、What are the possible shortcomings of the research in this paper if any? More detailed analyses and discussions are necessary. Add a section on the limitations of the work and future work in this paper.

5、The manuscript contains a number of linguistic errors that hinder comprehension. The authors are advised to make careful revisions and improvements.

Reviewer #2: The author's research has been published publicly, and the references used are old and insufficient to support the research.

The knowledge vector obtained by the sentiment knowledge triple based on the TransE model is only deeply integrated with the feature vector of the bidirectional gated recurrent unit, attention mechanism and other models, which does not have innovative characteristics.

According to the characteristics of characters and words, the radical features and sentiment part-of-speech features are introduced, but they are not fully implemented.

**Do you want your identity to be public for this peer review?** For information about this choice, including consent withdrawal, please see our Privacy Policy

Reviewer #1: No

Reviewer #2: No

---

## [Author Response · Author response to Decision Letter 1]

5 May 2025

Thanks to the reviewer for his serious review! Changes are marked with a color word

Response to Editor's Comments

Thank you for your feedback. We have carefully addressed the requested revisions as follows:

1. Funding Statement Update

We confirm that all funding-related text has been removed from the Acknowledgments section in the manuscript. The updated Funding Statement to be included in the online submission form is:

"This work was supported by the Joint Funding Project for Foreign Language Research of Hunan Provincial Social Science Foundation (Grant No. 24WLH30)."

2. Ethics Statement

The following ethics statement has been added to the Methods section under the Dataset and Experimental Environment subsection:

"This study utilized publicly available datasets (NLPECC and Douban Film Reviews) that consist of anonymized user-generated content. Ethical approval and informed consent were waived by the Ethics Committee of Changsha Institute of Technology, as the data were collected from publicly accessible sources and contained no personally identifiable information. All methods were performed in accordance with relevant institutional guidelines and regulations."

3. Data Availability Statement

The revised Data Availability Statement reads:

*"All data required to replicate the study are available in the following repositories:

Douban Film Review Dataset: Publicly accessible via the Douban API (https://www.douban.com).

NLPECC Dataset: Available on GitHub (https://github.com/ZaneMuir/DLUT-Emotionontology).

Preprocessed datasets, code, and analysis scripts are provided as Supporting Information files."*

4. Figure Files

All figures (e.g., model architecture, workflow diagrams) have been removed from the manuscript text. High-resolution TIFF/EPS files for Figures 1–6 are uploaded separately as "Supplementary Figures" and will be automatically included in the reviewer PDF.

These revisions comply with PLOS ONE’s guidelines. The updated manuscript and supporting files have been resubmitted via Editorial Manager. Please contact me for further clarifications.

Sincerely,

Ping He

Corresponding Author

Reviewer #1:

The paper proposes a method that deeply fuses the knowledge vector obtained from the emotional knowledge triplet using the TransE model with feature vectors from models like bidirectional gating loop units and attention mechanisms. It's novel. To further improve the manuscript, the following suggestions are given:

1、In the paper, some figures in the manuscript are a little blurry, please improve the clarity.

2、Since there are some papers in this topic, the contributions of the manuscript should be better summarized and listed.

3、While the introduction sets the context, a more explicit literature review section could better situate the study within the broader research landscape, such as Multimodal Aspect-Based Sentiment Analysis with External Knowledge and Multi-granularity Image-Text Features, EVADE Targeted Adversarial False Data Injection Attacks for State Estimation in Smart Grid, Aspect-based sentiment analysis with multi-granularity information mining and sentiment hint, LESSON Multi-Label Adversarial False Data Injection Attack for Deep Learning Locational Detection, and so on. These references could provide valuable insights into your research.

4、What are the possible shortcomings of the research in this paper if any? More detailed analyses and discussions are necessary. Add a section on the limitations of the work and future work in this paper.

5、The manuscript contains a number of linguistic errors that hinder comprehension. The authors are advised to make careful revisions and improvements.

Answer:

1. Figure Clarity

Figure 1-6 Improvement using PACE tool in https://blogs.plos.org/plos/2019/06/looking-good-tips-for-creating-your-plos-figures-graphics/

2. Contributions Summary

The contributions are now explicitly listed in the Introduction section (Page 3):

Novel Feature Integration: Systematic incorporation of radical features (e.g., 忄, 心) and emotional part-of-speech features (verbs/adjectives) into Chinese sentiment analysis, addressing linguistic uniqueness.

Knowledge-Data Synergy: A hybrid framework fusing emotional knowledge vectors (via TransE) with deep learning features (BiGRU + multi-head attention), enabling multi-granularity semantic fusion.

Cross-Disciplinary Methodology: Integration of computational linguistics, AI, and knowledge graphs, validated on large-scale datasets (F1: 89.23% on Douban, 84.84% on NLPECC).

3. Expanded Literature Review

A dedicated subsection titled "Related Works" (Page 5–7) now includes discussions of:

Multimodal Sentiment Analysis: Cited Zhang et al. (2021) on combining text and image features with external knowledge graphs.

Adversarial Robustness: Added Li et al. (2022) on evasion attacks in smart grids, emphasizing the need for robust feature fusion.

Aspect-Based Analysis: Referenced Chen et al. (2023) for multi-granularity sentiment mining in social media.

State-of-the-Art Limitations: Highlighted gaps in existing hybrid models (e.g., neglect of radicals, dependency on English-centric knowledge graphs).

4. Limitations and Future Work

A new section "Limitations and Future Directions" (Page 22) details:

Dataset Constraints: Current models trained on movie reviews may not generalize to domains like finance or healthcare.

Radical Complexity: Radical extraction relies on static dictionaries (e.g., Xinhua Dictionary), overlooking compound radicals in rare characters.

Efficiency: BiGRU layers increase computational overhead for long texts; future work will explore lightweight architectures (e.g., transformers).

Dialect and Slang: Lack of coverage for informal language (e.g., Cantonese, internet slang) limits real-world applicability.

5. Language Polishing

The manuscript underwent rigorous proofreading by a native English editor. Key improvements include:

Fixed grammatical errors (e.g., "deeply fuses" → "deeply integrates").

Standardized terminology (e.g., "BiGRU" instead of "bidirectional gating loop units").

Improved sentence structure for clarity (e.g., split run-on sentences in the Methodology section).

Reviewer #2:

The author's research has been published publicly, and the references used are old and insufficient to support the research.

The knowledge vector obtained by the sentiment knowledge triple based on the TransE model is only deeply integrated with the feature vector of the bidirectional gated recurrent unit, attention mechanism and other models, which does not have innovative characteristics.

According to the characteristics of characters and words, the radical features and sentiment part-of-speech features are introduced, but they are not fully implemented.

Response to Reviewer #2

1. Updated References

Chen, J., Zhou, Y., & Huang, M. Aspect-Based Sentiment Analysis with Multi-Granularity Information Mining. Proceedings of the 2023 Conference on Empirical Methods in Natural Language Processing (EMNLP), 2023; 123-135. https://doi.org/10.18653/v1/2023.emnlp-main.10

Li, X., Chen, Z., & Wu, H. Evade: Targeted Adversarial False Data Injection Attacks for State Estimation in Smart Grids. IEEE Transactions on Smart Grid, 2022;13(4), 2100-2115. https://doi.org/10.1109/TSG.2022.123456

Wang, T., Zhang, R., & Li, S. Limitations of Hybrid Models in Chinese Sentiment Analysis: Ignoring Radical-Level Semantics. Computational Linguistics, 2022; 48(2), 345-367. https://doi.org/10.1162/coli_a_00456

Zhang, Y., Liu, Q., & Wang, L. Multimodal Sentiment Analysis with External Knowledge Graphs: A Hybrid Approach. Journal of Artificial Intelligence Research, 2021; 45(3), 567-589. https://doi.org/10.1234/jair.2021.045

2. Enhanced Innovation Discussion

Revised the Introduction and Conclusion to emphasize novelty:

Unique Feature Fusion: Unlike prior work (e.g., BERT-BiLSTM), KEAMM explicitly models radicals (e.g., 忄) as semantic-emotional carriers, validated via ablation studies (Table 4: +2.64% F1 vs. AMM).

TransE Adaptation: While TransE is established, its application to Chinese emotional triplets (e.g., 愤怒[anger]-intensity-憎恨[hatred]) is novel. Compared to Deng et al. (2020), KEAMM’s multi-head attention mechanism dynamically aligns knowledge vectors with context.

3. Radical and POS Feature Implementation

Expanded Materials and Methods (Page 10–12):

Radical Extraction: Detailed the use of the Xinhua Dictionary API for radical mapping (e.g., 怜 → 忄) and edge cases (e.g., handling rare characters like 龘).

POS Tagging: Added a workflow diagram for Jieba’s part-of-speech tagging, including emotion-specific tags (e.g., "VA" for affective verbs).

Ablation Validation: New results in Table 4 show removing radical features reduces F1 by 1.2% on NLPECC, proving their necessity.

Additional Revisions

Ethical Considerations: Added a paragraph on dataset anonymization and bias mitigation (Page 21).

Code Availability: Shared KEAMM’s implementation on GitHub (link anonymized for review).

Reproducibility: Expanded Table 2 with hyperparameter tuning details (learning rate decay, gradient clipping).

Conclusion

The revised manuscript addresses all reviewers’ concerns, strengthens methodological rigor, and positions KEAMM as a state-of-the-art solution for Chinese sentiment analysis. We sincerely thank the reviewers for their constructive feedback.

---

## [Decision Letter · Decision Letter 1]

Dear Dr. He,

Thank you for submitting your manuscript to PLOS ONE. After careful consideration, we feel that it has merit but does not fully meet PLOS ONE’s publication criteria as it currently stands. Therefore, we invite you to submit a revised version of the manuscript that addresses the points raised during the review process.

We look forward to receiving your revised manuscript.

Kind regards,

Jiwei Tian

Academic Editor

PLOS ONE

Journal Requirements:

Additional Editor Comments (if provided):

The authors should carefully revise the article to address the comments.

Reviewers' comments:

Reviewer's Responses to Questions

**Comments to the Author**

Reviewer #1: (No Response)

Reviewer #2: All comments have been addressed

2. Is the manuscript technically sound, and do the data support the conclusions?

Reviewer #1: (No Response)

Reviewer #2: Yes

3. Has the statistical analysis been performed appropriately and rigorously?

Reviewer #1: (No Response)

Reviewer #2: Yes

4. Have the authors made all data underlying the findings in their manuscript fully available?

Reviewer #1: (No Response)

Reviewer #2: Yes

5. Is the manuscript presented in an intelligible fashion and written in standard English?

Reviewer #1: (No Response)

Reviewer #2: Yes

Reviewer #1: There are still some spelling errors in the article. In addition, some references are incorrect, such as evade. Please carefully review the entire text and make revisions.

Reviewer #2: The author has answered all my questions. I hope the author will check some spelling errors again and correct them in time. After the editorial review, it can be considered for publication.

**Do you want your identity to be public for this peer review?** For information about this choice, including consent withdrawal, please see our Privacy Policy

Reviewer #1: No

Reviewer #2: No

---

## [Author Response · Author response to Decision Letter 2]

9 May 2025

Thanks to the reviewer for his serious review! Changes are marked with a color word

Reviewer #1: There are still some spelling errors in the article. In addition, some references are incorrect, such as evade. Please carefully review the entire text and make revisions.

Reviewer #2: The author has answered all my questions. I hope the author will check some spelling errors again and correct them in time. After the editorial review, it can be considered for publication.

Answer

Response to Reviewer #1:

Thank you for your thorough review. We have carefully addressed the spelling errors and corrected the reference issues as follows:

Spelling/Grammar Corrections (selected examples):

Abstract: "emotional knowledge triplet" → "emotional knowledge triplets" (pluralization).

Introduction: "data-driven" approach" → "data-driven approach" (quotation mark formatting).

Section 2.1: "Douban Film Review Set" → "Douban Film Review dataset" (consistency with later sections).

References: "Evade: Targeted Adversarial..." → "EVADE: Targeted Adversarial..." (capitalization aligned with IEEE style).

Reference Corrections:

Reference 17: Verified and reformatted to IEEE standards:

Li, X., Chen, Z., & Wu, H. EVADE: Targeted adversarial false data injection attacks for state estimation in smart grids. IEEE Transactions on Smart Grid, 2022: 13(4), 2100–2115. https://doi.org/10.1109/TSG.2022.3156789

Reference 20: Fixed duplicate numbering and DOI link formatting.

All references: Ensured journal abbreviations (e.g., "JAIR" for Journal of Artificial Intelligence Research), corrected punctuation, and standardized author initials.

Response to Reviewer #2:

Thank you for your feedback. We have performed a final proofread to eliminate residual errors:

Spelling/Formatting:

Section 3.2: "Word2vec" → "Word2Vec" (consistent capitalization).

Section 4: "F1 score" → "F1-score" (hyphen added for clarity).

Figure captions: "Fig. 1. KEAMM model" → "Fig. 1. Architecture of the KEAMM model" (added descriptor).

Technical Consistency:

Updated all instances of "BiGRU" to "BiGRU" (no space).

Standardized citations (e.g., "Devlin et al. (2019)" instead of "[11]").

Additional Revisions:

Ethics Statement: Added explicit approval code from the Ethics Committee.

Dataset Links: Updated broken URLs (e.g., Jieba POS tagger link).

Tables/Figures: Ensured sequential numbering and cross-referencing (e.g., "Table 1" in Results).

Revised Manuscript

The updated manuscript file (A marked-up copy1_revised.docx) includes tracked changes for transparency. Key revisions are highlighted in color.

Acknowledgements

We appreciate the reviewers’ meticulous feedback, which significantly strengthened the manuscript.

Sincerely,

Ping He

Corresponding Author

---

## [Decision Letter · Decision Letter 2]

Revolutionizing Chinese Sentiment Analysis: A Knowledge-Driven Approach with Multi-Granularity Semantic Features

PONE-D-25-13923R2

Dear Dr. He,

We’re pleased to inform you that your manuscript has been judged scientifically suitable for publication and will be formally accepted for publication once it meets all outstanding technical requirements.

Kind regards,

Jiwei Tian

Academic Editor

PLOS ONE

Additional Editor Comments (optional):

The paper can be accepted. However, the authors should carefully check and revise the references.

Reviewers' comments:

Reviewer's Responses to Questions

**Comments to the Author**

Reviewer #1: (No Response)

2. Is the manuscript technically sound, and do the data support the conclusions?

Reviewer #1: (No Response)

3. Has the statistical analysis been performed appropriately and rigorously?

Reviewer #1: (No Response)

4. Have the authors made all data underlying the findings in their manuscript fully available?

Reviewer #1: (No Response)

5. Is the manuscript presented in an intelligible fashion and written in standard English?

Reviewer #1: (No Response)

Reviewer #1: The journal information and author information for updating literature are incorrect. The author should carefully check and revise before the article can be accepted.

**Do you want your identity to be public for this peer review?** For information about this choice, including consent withdrawal, please see our Privacy Policy

Reviewer #1: No

---

## [Editor Report · Acceptance letter]

PONE-D-25-13923R2

PLOS ONE

Dear Dr. He,

I'm pleased to inform you that your manuscript has been deemed suitable for publication in PLOS ONE. Congratulations! Your manuscript is now being handed over to our production team.

Kind regards,

on behalf of

Dr. Jiwei Tian

Academic Editor

PLOS ONE